# Protection of Whey Polypeptide on the Lipid Oxidation, Color, and Textural Stability of Frozen–Thawed Spanish Mackerel Surimi

**DOI:** 10.3390/foods12244464

**Published:** 2023-12-13

**Authors:** Yunying Li, Lingru Kong, Xiaotong Zhang, Rongxin Wen, Xinyan Peng

**Affiliations:** College of Life Sciences, Yantai University, Yantai 264005, China; 18596200343@163.com (Y.L.); konglr2226@163.com (L.K.); zhangxiaotong0622@163.com (X.Z.); 18800434580@163.com (R.W.)

**Keywords:** surimi, whey protein polypeptide, freeze–thaw stability, quality

## Abstract

Repeated freeze–thaw (FT) cycles can have an impact on surimi quality. In this study, we used 0.02% BHA as a positive control group. We examined the effects of different concentrations (0%, 5%, 10%, and 15%) of whey protein hydrolysate (WPH) on surimi, focusing on alterations in color metrics (L* for brightness, a* for red–green, b* for yellow–blue, and overall whiteness), textural characteristics, and antioxidant capacity during various freeze–thaw (FT) cycles. The results showed that the lipid oxidant values of surimi, as well as its a* and b* values, rose as the number of FT cycles increased; whereas the adhesiveness, resilience, gumminess, and shear force dropped, as did L* and the whiteness values, leading to an overall darkening of color and gloss. By contrast, the study found that the addition of WPH could effectively slow down the decrease of surimi textural stability after repeated freeze–thawing, with the textural stability of the group with 15% WPH being significantly superior to those of the other groups (*p* < 0.05). Under the same number of cycles, adding 15% WPH to the experimental group could successfully lower total volatile basic nitrogen (TVB-N) and effectively increase the antioxidant activity of surimi. This finding suggested that 15% WPH had the greatest effect on increasing surimi FT stability. To conclude, it was proved that WPH can be added to frozen surimi and improve its quality.

## 1. Introduction

The texture and nutritional content of surimi has helped it become more popular among consumers in recent years [1]. Smooth, and elastic surimi products especially those made from mackerel are low in cholesterol and high in biologically active substances like fat-soluble vitamins (E and D), readily digested proteins, fats, and vitamins, as well as essential long-chain omega-3 fatty acids, docosahexaenoic acid (DHA), eicosapentaenoic acid (EPA), and easily digestible fats [2]. These substances are vital to a human diet and have been demonstrated to help prevent many diseases [3]. After being produced, surimi is frequently frozen to secure its subsequent transit and sale and maintain its quality [4]. However, the frozen surimi must go through multiple freezing and thawing stages before consumption because the cold chain system is inadequate during production, storage, and marketing operations [5]. Cryoprotectants must be added to stabilize the surimi quality and texture and prevent water loss after repeated freezing and thawing [6]. The most popular way to effectively prevent freezing denaturation of surimi proteins at the moment is cryoprotectants [7]. Zhang et al. [6] discovered that surimi byproduct hydrolysates effectively delay lipid and protein oxidative denaturation in surimi during freezing, demonstrating their potential as cryoprotectants.

The protective agents represented by whey proteins are byproducts extracted from cheese [8]. Whey proteins are valuable and nutritional, with considerable branched-chain amino acids and abundant bioactive molecules [9]. Simultaneously, studies have shown that whey protein has particular antioxidant activity, among which α-lactalbumin and β-lactoglobulin can inhibit fat oxidation by scavenging free radicals [10]. Whey protein hydrolysis (WPH) has been used in many situations and has been shown to have antioxidant, anti-inflammatory [11], anti-obesity, and anti-hypertensive properties [12]. Among them, WPH antioxidant activity is a significant biological benefit and has recently attracted the attention of experts. Notably, some peptides with antioxidant activity have been said to have a great deal of potential as natural antioxidants in food products [13]. Furthermore, there is proof that adding antioxidant peptides can improve food product stability and physicochemical characteristics [14].

Previous research revealed that WPH has strong antioxidant capabilities in the growth of Arctic charr [15] and also demonstrated good hygroscopicity due to its hydrolysis degree ranging between 13.82% and 15.35% [16]. However, it is still unknown how WPH protects surimi product color and textural integrity. In this study, whey polypeptides were incorporated into surimi that had undergone multiple freezing and thawing cycles, aiming to investigate their effects on the quality of the surimi. This was done to further validate the improvement mechanisms of WPH on the quality of meat products during storage processes. In the current investigation, surimi that had been repeatedly frozen and thawed was mixed with various concentrations of WPH, with 0.02% butyl hydroxyanisole (BHA) serving as a positive control. This study aimed to investigate the effect of WPH on the quality characteristics of frozen–thawed surimi by measuring changes in antioxidant activity (thiobarbituric acid reactive substances and total volatile basic nitrogen), color difference (L*, a*, and b*), and textural properties (gumminess stability, resilience stability, adhesiveness stability, and shear force stability) during the freeze–thawing processes. Importantly, this will lay the groundwork for the theoretical application of WPH in surimi products and provide a basis for its eventual large-scale industrial application.

## 2. Materials and Methods

### 2.1. Chemicals and Materials

Yanda Market provided AA-grade fresh *Scomberomorus niphoniu*. The supplier of whey protein, Beijing Milky Way Commercial Company, provided a protein level of purity of almost 95%. Alkaline protease (6 × 10^4^ U/g) was obtained from the Novo Company in Denmark for use in the experiment. Furthermore, the Sigma Company in the US was used to obtain BHA. In this experiment, only analytical-grade chemicals were employed.

### 2.2. Preparation of WPH

WPH was created using the Peng et al. [17] approach with the necessary changes. Natural whey peptide (NWP) was warmed at 95 °C for five minutes before compounding into a 5% solution. Alkaline protease was added and hydrolyzed for 5 min with a protein-to-enzyme (E/S) ratio of 2:100 after cooling to 65 °C. Following this, a washing and distillation procedure was carried out, in which a 1 mol/L NaOH solution was continually added to the hydrolysate to maintain its pH at 8.0. Following washing, the enzyme was entirely deactivated by 5 min of boiling in a water bath. Furthermore, the hydrolysate was freeze-dried after cooling to room temperature to produce WPH for further experimental components.

### 2.3. Preparation of Surimi

The surimi was made using the Sun et al. [18] approach with the necessary modifications. According to the State Council of China’s regulations from 1988 and the “Guidelines for the Treatment of Experimental Animals” published by the Chinese Ministry of Science and Technology in 2006, *Scomberomorus niphonius* fish were physically beaten on the head before having their internal organs removed and cleaned. After the fish had been cleaned and cut into small pieces, the heads and bones were removed, and the surimi was collected using a meat separator (HZ250, Yingbo Machinery Co., Ltd., Xiamen, China). The surimi was blended in an ice bath for 2 min using a homogenizer, the Ultra Turrax from IKA in Germany. Before the moisture content of the mixture reached 80%, the moisture content was determined using a standard drying oven (AOAC, 1990), and no additives were added to the control group. However, the other five groups received various concentrations of WPH at 5%, 10%, and 15%, NWP at 10%, BHA at 0.02%, and ice-distilled water. The experiment was designed so that each group had a unique combination of these additives. The mixture was stirred well for 10 min in an ice bath below 4 °C. The resulting fish paste was transferred into a 15 mL centrifuge tube and spun at 1500× *g* for 5 min to eliminate air bubbles. Each surimi sample underwent a comprehensive freeze–thaw cycle, completely freezing at −20 °C for 20 h, followed by a thawing process at 4 °C for 12 h until the core temperature reached between 0 °C and 2 °C. This process constituted a single freeze–thaw cycle. The stability of the surimi under freeze–thaw conditions was then evaluated after 0, 1, 3, 5, and 7 such cycles, adhering to the procedures previously outlined.

### 2.4. The Measurement of Thiobarbituric Acid Reactive Substances

Thiobarbituric acid reactive substances (TBARS) were measured according to Ghimire et al. [19] with some modifications. In brief, each group was sampled from surimi thawed at different cycles and equilibrated at room temperature. The surimi (10.0 g) was mixed with 25 mL of trichloroacetic acid (TCA) solution and 10 mL of distilled water in this test. Then, the homogenate was centrifuged at 2000× *g* for 15 min to obtain the supernatant. Subsequently, 4 mL of the supernatant was mixed with thiobarbituric acid solution and heated in boiling water for 15 min. A spectrophotometer measured the solution’s absorbance at 532 nm after it had cooled to room temperature. Malonaldehyde was then used as the standard. Results were expressed as mg malonaldehyde equivalents per kg surimi (MDA/kg). The computation was expressed using the following formula [20]:TBARS (mg/kg) = (A_532_/W_s_) × 9.48(1)

Within this analytical framework, the variable A_532_ is defined as the optical density quantified at a wavelength of 532 nm, reflecting the absorptive capacity of the solution. At the same time, W_s_ represents the mass of the carnivorous sample in grams. The constant 9.48 is extrapolated from the molar extinction coefficient, quantified as 152,000 M^−1^ cm^−1^, in conjunction with the dilution coefficient relevant to the colorimetric assay of thiobarbituric acid reactive substances.

### 2.5. The Measurement of Total Volatile Basic Nitrogen

A Kjeldahl apparatus was used to determine total volatile basic nitrogen (TVB-N) [21]. The microtitration method was utilized to measure the TVB-N value of the *Scomberomorus niphonius* fish surimi, and mg N/100 g was used to express the TVB-N value. 

### 2.6. The Measurement of Gumminess

With the necessary adjustments, texture profile analysis (TPA) was carried out using the modified techniques of Yang et al. [22]. Surimi that had been thawed and subjected to various numbers of FT cycles was used to collect samples. The samples were cut into cylinders 15 mm in height and 17 mm in diameter from the center for texture measurement. A flat-ended plunger with a P/50 (5 cm diameter) was used with a texture analyzer (TA-XT. Plus, Stable Micro Systems, Godalming, UK). With compression and return speeds of 1.00 mm/s, respectively, the TPA settings were set to compression mode. The interval between the two compression test cycles was 5.0 s, and the trigger force was 10 g. The pre-test, detection, and post-test speeds were set as 1.00 mm/s. Each group required at least 6 repeated tests, and the TPA results were recorded as gumminess.

### 2.7. Determination of Resilience

The resilience of the surimi was determined with appropriate modifications according to the method of Wang et al. [23]. Samples (2.0 cm × 4.0 cm × 2.0 cm) were taken from surimi that had undergone different FT cycles and thawed. After the surimi was cooled to room temperature, a texture analyzer (TA-XT. Plus, Stable Micro Systems, Godalming, UK) was used for two-cycle compression testing (analytical compression trials were executed on an identical batch of specimens during two separate, defined test intervals, with each interval dedicated to the examination of a unique specimen). The TPA parameters were set as follows: the time interval between two-cycle compression tests was 5.0 s, P/50 flat cylindrical probe (diameter 5 cm), probe test speed was 5.00 mm/s crosshead speed, each group of samples was tested 6 times, and the TPA result was recorded as resilience.

### 2.8. Determination of Adhesiveness

Using the technique described by Zhang et al. [24] with the necessary adjustments, texture analysis of the surimi was carried out using a texture analyzer (TA-XT Plus, Stable Micro Systems Ltd., Godalming, UK). After the samples were cut into 1 cm × 1 cm × 1 cm, the two-cycle compression test with a time interval of 5.0 s, a P/50 cylindrical probe, a test speed of 1.00 mm/s, and a trigger force of 5.0 g were the settings for the TPA. The TPA findings were recorded as adhesiveness, and each sample was tested 6 times for replicates.

### 2.9. Determination of WPH on the Color of Surimi

Surimi goods were used to test the colors. The approaches of Gan et al. [20] and Hwang et al. [21] outlined were used to determine the L*, a*, and b* values in surimi products. Samples were taken from surimi that had undergone different FT cycles. A chromameter (CR-400, Konica Minolta, Tokyo, Japan) was used to measure and record the values of L*, a*, and b* after each sample was uniformly sliced into 20 mm lengths. Every parameter measurement is subjected to three independent repeats. The average value of the three experiments is then calculated by summing the results of these three replicates and dividing the result by three.

The formula for calculating whiteness value [25] is as follows:(2)$$W^{2}=100-\sqrt{{\left(100-L^{*}\right)}^{2}+{\left(a^{*}\right)}^{2}+{\left(b^{*}\right)}^{2}}$$

### 2.10. Determination of Shear Force

Shear force determination method: Following the research methods of Kawamura et al. [26] and Shi et al. [27], appropriate modifications were made to conduct shear force measurements on surimi. Each batch of samples was put into an 80 °C water bath after going through various freezing and thawing cycles. The samples were taken out and chilled to room temperature (25 °C) once the internal temperatures reached 70 °C as determined by a thermometer. The shear force of the surimi samples was measured using a muscle tenderization instrument (MAQC-12, Nanjing Mingao Instrument Equipment Co., Ltd., Nanjing, China). Each group was measured six times to obtain an average result.

### 2.11. Statistical Analysis

Variance analysis and statistical analysis were used as part of a methodological approach to assess the effects of different dose treatments and FT cycles on surimi quality indicators. Statistix 8.1, a program developed by Analytical Software, St. Paul, MN, USA, was used for all statistical calculations. We used Tukey’s multiple comparisons along with analysis of variance (ANOVA) to determine the significance of the treatment effects (*p* < 0.05). The experiments were conducted in triplicate, each replicated three times. The standard deviation is displayed alongside the mean values in the data.

## 3. Results and Discussion

### 3.1. Effect of WPH on TBARS Values of Repeated Freeze–Thawing Surimi

The TBARS value is an important biomarker for detecting the freshness quality of meat and is used to determine the degree of fat oxidation [28]. Figure 1 depicts the TBARS values of surimi samples in various groups during the FT cycles. The TBARS values rapidly increased as the number of FT cycles increased. This phenomenon could be caused by the deterioration of muscle cell integrity, resulting in the release of lipase, which ultimately promotes lipid oxidation and causes an increase in TBARS values [29]. Rahmanifarah et al. [30] discovered that frozen storage may release free fatty acids, which can cause lipid oxidation in aquatic foods. At the beginning of the FT cycle, the TBARS values of the six groups ranged from 0.045 to 0.049. However, there was no significant difference (*p* > 0.05). After seven FT cycles, the TBARS values in the blank control group, 10% NWP, 5% WPH, 10%WPH, 15%WPH, and 0.02%BHA increased by 262%, 244%, 239%, 213%, 155%, and 175%, respectively. These findings support a prior study that found that adding WPH during the freezing process can effectively limit the oxidation of myofibrillar protein in meat [31]. Moreover, the peptides appeared to have a vital purpose in reducing meat fat oxidation. Wang et al. [32] discovered that potato hydrolyzed polypeptides can lower TBARS values in pork. These findings suggested peptides, including WPH, may successfully inhibit surimi oxidation during repeated FT procedures. Furthermore, the 15% WPH addition group had the lowest TBARS value among these groups in the first, third, fifth, and seventh FT cycles (*p* < 0.05), and the effect was comparable to the 0.02% BHA addition group.

### 3.2. Effect of WPH on TVB-N Values of Repeated Freeze–Thawing Surimi

TVB-N values are frequently used to indicate the degree of amine and protein degradation, which is a critical quality evaluation for identifying the freshness of surimi products [33]. Figure 2 depicts the effect of WPH addition on the TVB-N content of surimi during FT cycles. The TVB-N content of surimi increased with increasing FT numbers in all groups. This could be related to the degradation of proteins and other nitrogen-containing substances caused by enzyme-induced spoilage during the FT cycles, resulting in the accumulation of organic amines, as seen by an increase in TVB-N [34]. Notably, Zong et al. [35] discovered that the TVB-N value of a large yellow croaker gradually increased during frozen storage. The TVB-N values of surimi in this study were 7.14–7.47 at 0 cycles and then increased in all groups as the FT numbers increased. The TVB-N values in the blank control group increased rapidly from 7.36 (0 cycles) to 28.98 (7th cycle), substantially greater than the other five groups (*p* < 0.05). Interestingly, WPH significantly reduced the growth of TVB-N, particularly in the high WPH addition group (15%). This is consistent with the information noted in Section 3.1 of the study, which revealed that WPH addition had a beneficial influence on surimi freshness. When the fish paste is fresh, it is less prone to deterioration, which slows down protein degradation and the production of ammoniacal amines and other basic nitrogenous substances in the fish paste, decreasing TVB-N values. Furthermore, after the third and fifth FT cycles, the TVB-N values in the 15% WPH and 0.02% BHA groups were significantly lower than those in the blank control group, 10% NWP, 5% WPH, and 10% WPH (*p* < 0.05). Tyrosine and methionine are less likely to be degraded in protein when the TVB-N value is lower, improving the protection of the nutrients [36]. In light of these findings, it can be concluded that the WPH addition demonstrated less protein degradation and nutritional loss than the blank control group. Moreover, the 15% WPH addition better regulated the TVB-N level of surimi in comparison to the other groups.

### 3.3. Effects of WPH on Gumminess Stability of Repeatedly Freeze–Thawing Surimi

Gumminess is an important indicator that dramatically impacts the mouthfeel and texture of food, which in turn has a major impact on how well customers accept the product [37]. Figure 3 displays the results of the gumminess of surimi treated with various additives after 0, 1, 3, 5, and 7 FT cycles. There was no discernible difference in gumminess between these groups when the samples were not put through FT cycles (*p* > 0.05). The gumminess of each group during the repeated FT processes started to stabilize and subsequently began to decline. This could be attributed to the formation and growth of ice crystals with an increase in the number of FT cycles, along with the disruption of the network structure brought on by recrystallization [38], also contributing to the breakdown of the interfacial film on the surimi’s surface and the decline in interfacial stability [39]. In addition, the development of ice crystals during freezing caused water in the surimi to be lost and redistributed, weakening the connections between hydrophobic molecules and further reducing gumminess [40]. The blank control group showed the greatest loss of gumminess, with a total loss of 148.76 g, likely caused by the significant destruction of the myofibrillar protein’s original structure in the unprotected surimi due to elevated FT cycles [41]. Only a decrease of 88.93 g and no discernible change were observed in the 15% WPH group’s gumminess, which remained stable. After seven FT cycles, the 15% WPH-added group had significantly more gumminess than the other treatment groups (*p* < 0.05). After 7 FT cycles, each group’s gumminess decreased by 20.82%, 18.63%, 19.14%, 15.70%, 12.53%, and 14.66% in comparison to fresh surimi, showing that gumminess decreased as the number of FT cycles increased. The order of the strength of gumminess of hairtail surimi was as follows: 15% WPH group > 0.02% BHA group > 10% WPH group > 10% NWP group > 5% WPH group > blank control group. The findings of the study of Sousa et al. [42] on the oxidative stability of green weakfish (*Cynoscion virescens*) byproduct surimi and surimi enhanced with a *Spondias mombin* L. waste phenolic-rich extract during cold storage were similar to this study’s findings. The results show that WPH and BHA can significantly reduce the gumminess of surimi throughout repeated FT cycles, with the 15% WPH group showing the best results.

### 3.4. Effects of WPH on the Resilience Stability of Repeatedly Freeze–Thawing Surimi

The ability of meat to recover after deformation under the same speed and pressure conditions is measured by its resilience, a crucial indicator for evaluating textural properties [43]. According to Figure 4, the experiment investigated how the resilience index of surimi changed after 0, 1, 3, 5, and 7 FT cycles under various additive treatments (blank, NWP, and WPH). Each group’s surimi became less resilient after undergoing a different number of FT cycles, likely due to the possibility that FT cycles can lead to conformational changes that are oxidation-driven, destroy protein molecular structure, and create an imbalance in the interactions between protein and water [44]. It could also be attributed to the frequent passage of the sample through the maximum ice crystal formation zone, resulting in relatively large ice crystals [45], the gradual destruction of the network, the ensuing growth of larger pores [46], and irreparable harm to the surimi tissue structure and reduction in its resilience. The effect of the WPH groups was the most significant (*p* < 0.05) throughout each freeze–thaw cycle, showing that WPH could increase the resilience stability of surimi during multiple FT cycles. Among them, the resilience of the treatment groups was higher than that of the blank control group. Moreover, the resilience reduction of each group eventually grew as the number of FT cycles rose. After seven iterations of the FT cycle, the resilience in blank control, 10% NWP, 5% WPH, 10% WPH, 15% WPH, and 0.02% BHA dropped by 8.41%, 10%, 11.57%, 9.79%, 6.34%, and 8.80%, respectively, in comparison to fresh surimi. This is due to the fact that when the number of FT cycles rises, the myofibrillar protein’s degree of denaturation deepens [45], the sample’s dehydration shrinkage trend rises, and the resilience declines [46]. Furthermore, the 15% WPH group had the highest level of resilience compared to the other groups (*p* < 0.05). Additionally, when combined with the findings of Section 3.3, it is clear that adding 15% WPH significantly reduces the loss of surimi gumminess properties during repeated FT cycles, preserves its structural stability, and lessens the mechanical damage that ice crystal formation during the FT process causes to muscle tissue. Notably, this improves recoverability in resilience stability studies.

These findings are comparable to the findings of Sousa et al. [42] about using *Spondias mombin* L. waste phenolic extract to improve the oxidative stability of *Cynoscion virescens* byproduct surimi and surimi during refrigeration. Notably, the effect of WPH is greatest in the 15% WPH group and successfully addresses the problem wherein surimi resilience declines as the frequency of repeated FT cycles increases.

### 3.5. Effects of WPH on the Adhesiveness Stability of Repeatedly Freeze–Thawing Surimi

A textural characteristic that reflects the quality of surimi is adhesiveness [47]. After 0, 1, 3, 5, and 7 FT cycles, the adhesiveness of surimi under various additional treatments was assessed, and the findings are shown in Figure 5. According to the graph, there were no notable variations in adhesiveness across the groups prior to the freezing treatment (*p* > 0.05). Adhesiveness of all groups diminished after one freeze–thaw cycle, probably due to the muscle fiber structure’s repeated mechanical injury brought on by ice crystal formation during the freeze–thaw process [48]. In addition, intracellular ice crystals may cause the denaturation of proteins with a resulting decline in adhesiveness [49]. Notably, the WPH and BHA groups showed considerably higher adhesiveness than the blank control group (*p* < 0.05), although the adhesiveness drop was most pronounced in the blank control group. Due to the hydrolysis of structural proteins and the destruction of muscle fiber [48] and membrane components of fat cells brought on by ice crystal formation, the adhesiveness of all groups steadily decreased as the number of FT cycles increased [49]. The treatment groups also displayed considerably higher adhesiveness than the blank control group (*p* < 0.05), with the 15% WPH group exhibiting the most considerable improvement. The control group demonstrated the fastest rate of adhesiveness reduction, particularly after the 5th and 7th FT cycles. This is consistent with the results of Zhang et al. [50], who found that substituting oleogel-in-water Pickering emulsion for pork backfat improved the freeze–thaw stability of pork sausage. The findings show that surimi adhesiveness steadily diminishes as the number of FT cycles increases. However, in line with the findings in Section 3.3 and Section 3.4, WPH and BHA can significantly improve adhesiveness and its characteristics. The improved adhesiveness stability of surimi is specifically caused by the augmentation in adhesive stability and resilience stability throughout the freezing cycle, with 15% WPH having the most noticeable effect.

### 3.6. Effects of WPH on the L*, a*, and b* Stability of Repeatedly Freeze–Thawing Surimi

L*, a*, and b* are significant reference indicators for the color of surimi products because color directly influences consumer acceptance of surimi products. Studies by Sriket et al. [51] and Tseng et al. [52] suggest that FT cycles can cause the hydrolysis and oxidation of proteins and lipids in meat, changing its color and appearance. This work used several additive treatments to examine how the L*, a*, and b* values of surimi change after 0, 1, 3, 5, and 7 FT cycles. As shown in Figure 6, before the samples were frozen, there were statistically significant differences between the 15% WPH group and the blank control group in the L* value experimental treatment group (*p* < 0.05). Similarly, in the b* value experimental treatment group, there were significant differences between the 0.02% BHA group and other treatment groups (*p* < 0.05). Notably, the a* value of the 15% WPH group was significantly higher than that of the 0.02% BHA group (*p* < 0.05). After various FT cycles, the blank control and treatment groups’ L* values dropped while their a* and b* values rose. This is explained by the browning reaction brought on by repeated freezing and thawing cycles, which allows moisture from surimi product cells to diffuse to the surface [53]. These findings align with those of Pan et al. [54], who investigated the impact of FT cycles on the quality of quick-frozen pork patties with various fat concentrations. The a* and b* values of the WPH and BHA groups were considerably lower than those of the blank control group after seven FT cycles (*p* < 0.05), with the lowest values being seen in the 15% WPH group. Additionally, the L* values of nearly every treatment group in the L* value experimental treatment group were greater than those of the blank control group, with the L* value of the 15% WPH group being substantially higher than that of the other groups (*p* < 0.05). From the findings described in Section 3.1 and Section 3.2, adding WPH during the FT cycle reduced the TBARS and TVB-N indices in surimi, slowed oxidative reactions, and prevented the buildup of tyrosine and lysine within proteins, which, when proteins aggregated, could result in a darkening of the surimi color. In turn, this resulted in lower a* and b* values when compared to the control group, while the L* value was higher. These findings suggest that WPH and BHA can both lower the a* and b* values of surimi products while raising the L* value during repeated FT cycles. Notably, adding 15% WPH had the greatest impact on all of them and outperformed the 0.02% BHA positive control group.

### 3.7. Effects of WPH on the Whiteness Stability of Repeatedly Freeze–Thawing Surimi

Intuitively representing the quality of surimi, whiteness is a thorough evaluation index of the color stability of surimi and also an essential reference factor influencing consumer approval of the product [55]. According to Figure 7, the experiment assessed how the whiteness values of surimi changed after 0, 1, 3, 5, and 7 FT cycles under various additive treatments. Only the 10% WPH treatment group findings before freezing in the whiteness experimental treatment group were statistically different from those of the other treatment groups (*p* < 0.05). The whiteness values of the groups containing 5% WPH, 10% WPH, and 0.02% BHA were lower than those of the blank control group following one freeze–thaw cycle (*p* < 0.05), presumably as a result of the light-yellow color of WPH and BHA. All group whiteness values declined as the number of FT cycles increased, mainly because whiteness is correlated with the degree of protein oxidation and denaturation [56]. Protein denaturation brought on by FT cycles results in the loss of free water from the surimi network structure, weakening the surimi ability to reflect light [5] and lowering the whiteness values. In addition, non-enzymatic interactions between protein amide groups and lipid oxidation may contribute to the decrease in the whiteness of the surimi during FT cycles [57]. After seven consecutive FT cycles, the blank control group showed the most notable fall in whiteness values, with a total loss of 7.41%. After the fifth and seventh FT cycles, when the 15% WPH group considerably outperformed the other groups in terms of whiteness values (*p* < 0.05), the surimi whiteness value became more sensitive to the WPH concentration. With only a 2.93% reduction after the seventh FT cycle, the whiteness value of the 15% WPH group was substantially higher than that of the other groups (*p* < 0.05). This is likely due to the WPH ability to prevent non-enzymatic browning, which is brought on by interactions between carbonyl compounds formed during FT cycles and the side chains of amino acids [58]. This is also comparable to the findings of the investigation of Wang et al [59] into the effects of phenolic compound pterostilbene on the oxidative stability and gelation characteristics of myofibrillar protein from chicken breast during post mortem frozen storage. The previously reported results show that when FT cycles increase, surimi whiteness decreases. Moreover, the addition of WPH can successfully lessen surimi oxidation, lowering the surimi TBARS and TVB-N values, influencing its L*, a*, and b* values, and slowing down the surimi general loss in whiteness. Thus, it is evident that WPH significantly improves the whiteness of surimi during repeated FT procedures. The effect is most noticeable when WPH is present in concentrations of 15% or higher, and it somewhat outperforms the effects of BHA.

### 3.8. Effects of WPH on Shear Force Stability of Repeatedly Freeze–Thawing Surimi

Shear force is a crucial indicator of meat’s freshness and tenderness [60]. Figure 8 depicts the outcomes of shear force in this experiment on surimi exposed to seven cycles of freezing and thawing under various additional treatments. Before the samples were frozen, adding 15% WPH increased the shear force of the surimi. In contrast, adding 10% NWP, 5% WPH, 10% WPH, or 0.02% BHA had no significant impact on shear force (*p* > 0.05). The shear force of the 15% WPH group did not considerably diminish until after the fifth FT cycle. However, the shear force of the blank control group did so after just one cycle. This decline may be brought on by the breaking down of myofibrillar proteins in surimi due to the ice crystals repeatedly forming and melting [61]. In particular, the shear force of the control group reduced by 61.42% compared to fresh surimi after seven cycles of freezing and thawing, whereas the 15% WPH group only decreased by 15.73%. Shear force was reduced in the treatment groups containing 10% NWP, 5% WPH, 10% WPH, and 0.02% BHA by 61.79%, 41.38%, 22.23%, and 39.22%, respectively. The surimi shear force results are applied in the following order: 15% WPH group is superior to the 10% WPH group, 0.02 percent BHA group, 5% WPH group, the control group, and 10% NWP group. These findings are consistent with those of Ali et al. [62], who found that increasing freezing and thawing cycles significantly reduced chicken breast meat’s shear force. Notably, the experimental results support the effectiveness of both WPH and BHA in preventing the shear force reduction during subsequent FT cycles of surimi. The study’s findings corroborate the judgments made in Section 3.3, Section 3.4 and Section 3.5. Surimi gumminess stability, resilience stability, and adhesiveness stability can all be improved by WPH. Since gumminess, robustness, and adhesiveness affect shear force, WPH further improves the surimi shear force stability. Interestingly, using 15% WPH yields the most notable effect, outperforming the effect of BHA.

## 4. Conclusions

TBARS, TVB-N, adhesiveness, resilience, gumminess, L* value, a* value, b* value, whiteness value, and shear force were among the texture indicators that were measured in this experiment on surimi treated with various FT cycles and additives (NWP, WPH, and BHA). The oxidation capacity of surimi increased following several FT cycles. However, its adhesiveness, resilience, gumminess, and shear force were reduced. Additionally, its a* and b* values increased, its L* and whiteness values declined, and its overall color darkened. Moreover, the textural stability of surimi was reduced more drastically as the FT cycles increased. However, inclusion of WPH can significantly slow down the deterioration of surimi textural stability following numerous FT cycles. The group that received a 15% WPH addition exhibited notably greater texture stability than the others. In conclusion, the best way to increase surimi freeze–thaw stability is to add 15% WPH. Consequently, WPH can be added to frozen surimi to improve its quality. This study could support the use of WPH in the food business to raise the standard of frozen meat.

## Figures and Tables

**Figure 1 foods-12-04464-f001:**
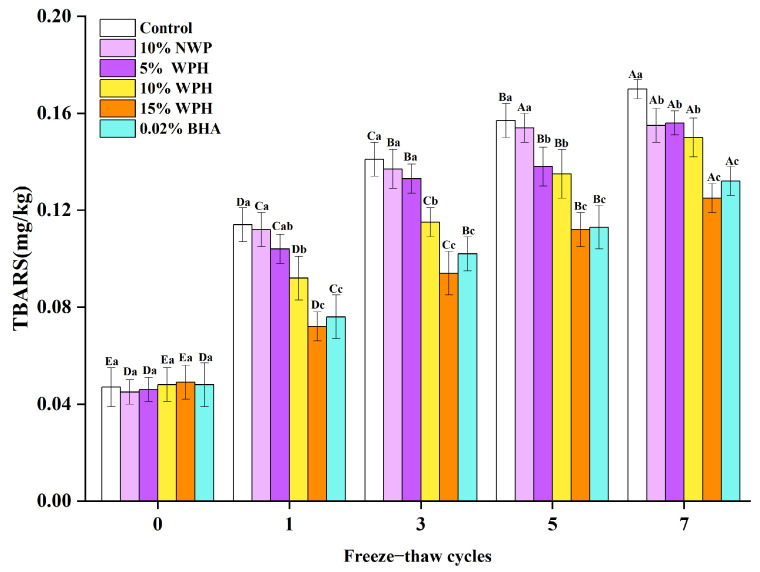
Effect of WPH on TBARS values of repeated FT surimi. Significant differences among different concentrations of WPH samples under the same number of FT cycles are indicated by different lowercase letters (a–c). Significant differences among the same samples with different numbers of FT cycles are denoted by uppercase letters (A–E). Control: no additives in the sample; NWP: native whey protein isolate; WPH: whey protein hydrolysates; BHA: butylated hydroxyanisole.

**Figure 2 foods-12-04464-f002:**
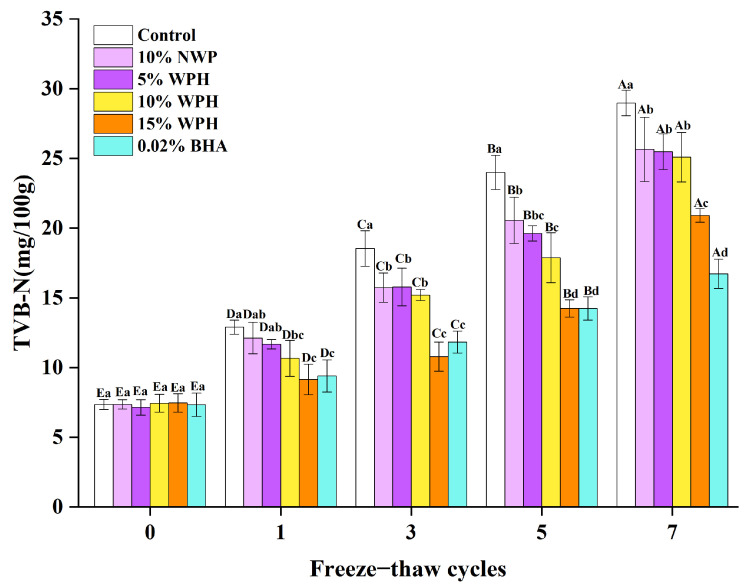
Effect of WPH on TVB-N values of repeated FT surimi. Significant differences among different concentrations of WPH samples under the same number of FT cycles are indicated by different lowercase letters (a–d). Significant differences among the same samples with different numbers of FT cycles are denoted by uppercase letters (A–E). Control: no additives in the sample; NWP: native whey protein isolate; WPH: whey protein hydrolysates; BHA: butylated hydroxyanisole.

**Figure 3 foods-12-04464-f003:**
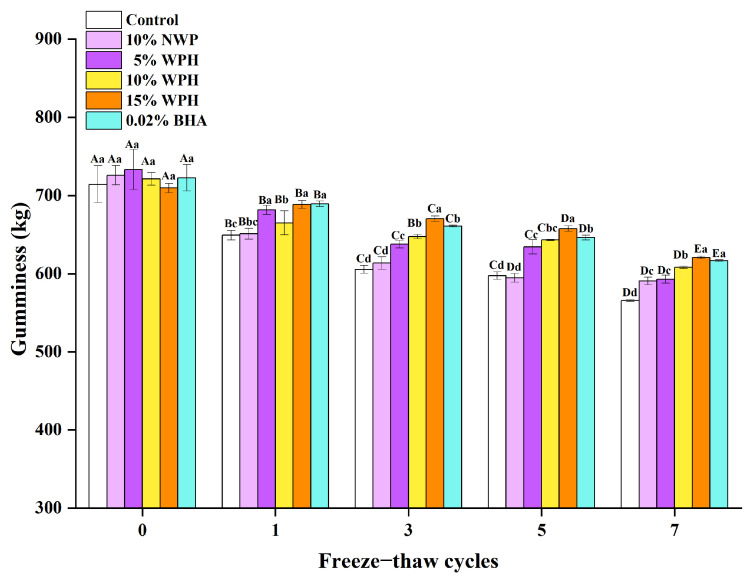
Effects of WPH on gumminess stability of repeatedly freeze–thawing surimi. Significant differences among different concentrations of WPH samples under the same number of FT cycles are indicated by different lowercase letters (a–d). Significant differences among the same samples with different numbers of FT cycles are denoted by uppercase letters (A–E). Control: no additives in the sample; NWP: native whey protein isolate; WPH: whey protein hydrolysates; BHA: butylated hydroxyanisole.

**Figure 4 foods-12-04464-f004:**
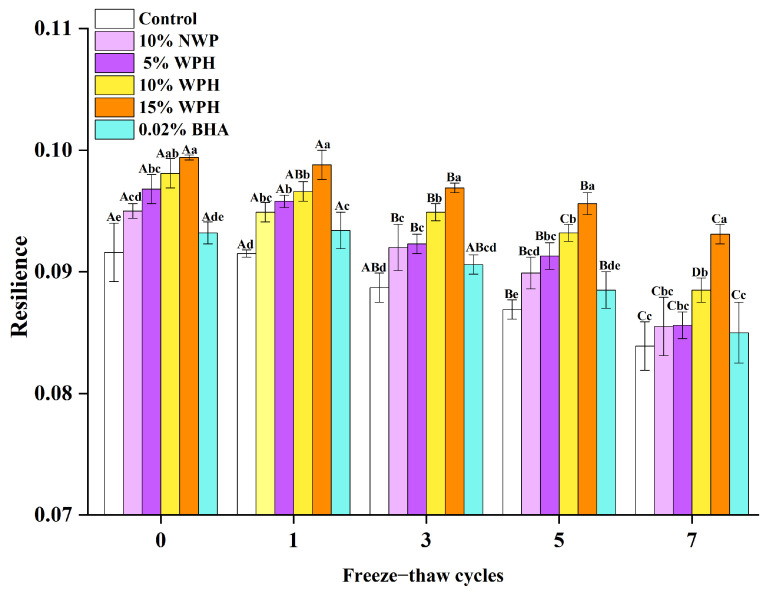
Effects of WPH on the resilience stability of repeatedly freeze–thawing surimi. Significant differences among different concentrations of WPH samples under the same number of FT cycles are indicated by different lowercase letters (a–e). Significant differences among the same samples with different numbers of FT cycles are denoted by uppercase letters (A–D). Control: no additives in the sample; NWP: native whey protein isolate; WPH: whey protein hydrolysates; BHA: butylated hydroxyanisole.

**Figure 5 foods-12-04464-f005:**
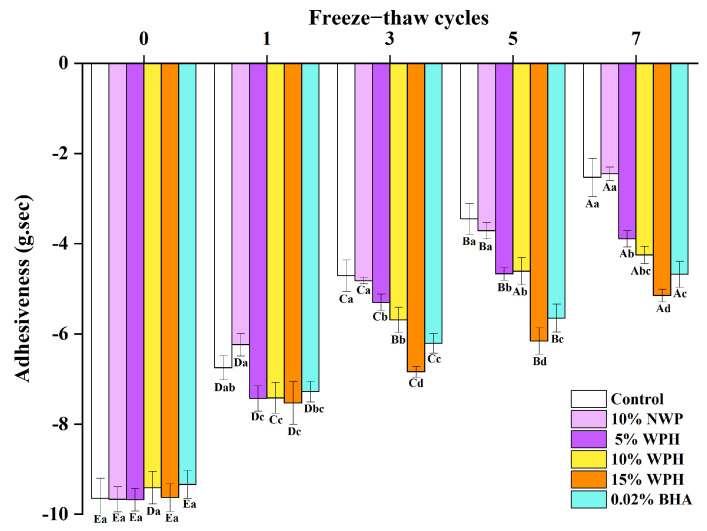
Effects of WPH on the adhesiveness stability of repeatedly freeze–thawing surimi Significant differences among different concentrations of WPH samples under the same number of FT cycles are indicated by different lowercase letters (a–d). Significant differences among the same samples with different numbers of FT cycles are denoted by uppercase letters (A–E). Control: no additives in the sample; NWP: native whey protein isolate; WPH: whey protein hydrolysates; BHA: butylated hydroxyanisole.

**Figure 6 foods-12-04464-f006:**
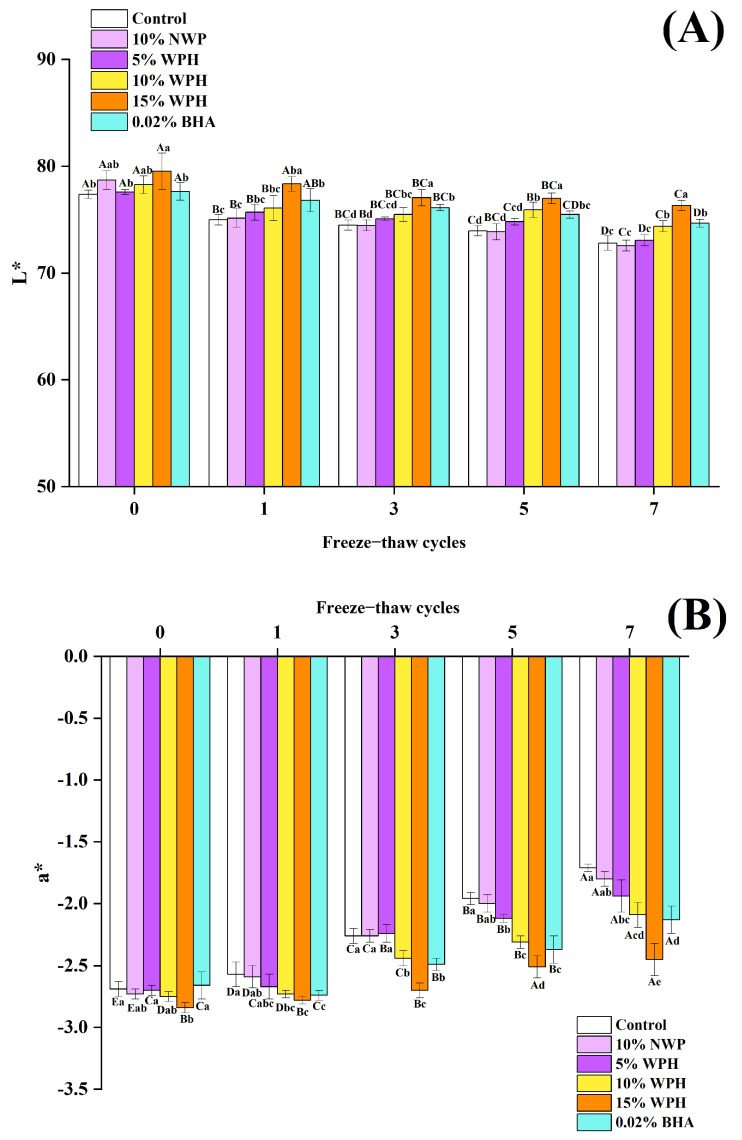
Effects of WPH on the L* (**A**), a* (**B**), and b* (**C**) stability of repeatedly freeze–thawing surimi. Significant differences among different concentrations of WPH samples under the same number of FT cycles are indicated by different lowercase letters (a–d). Significant differences among the same samples with different numbers of FT cycles are denoted by uppercase letters (A–E). Control: no additives in the sample; NWP: native whey protein isolate; WPH: whey protein hydrolysates; BHA: butylated hydroxyanisole.

**Figure 7 foods-12-04464-f007:**
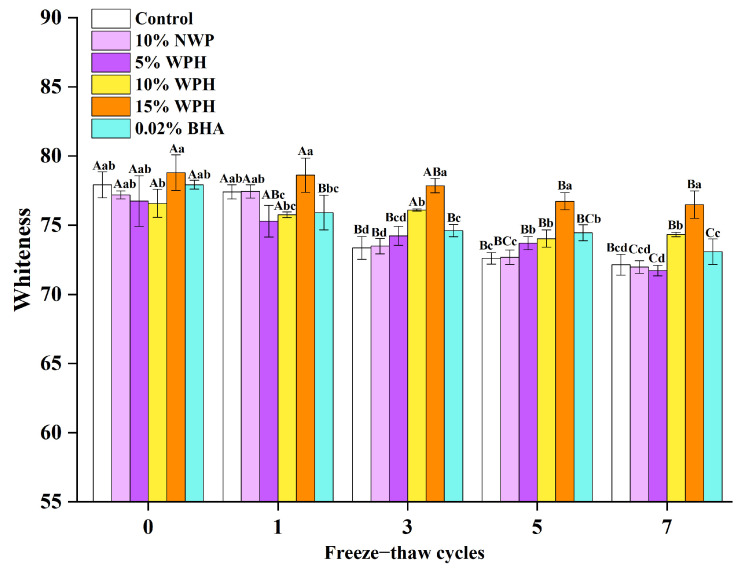
Effects of WPH on the whiteness stability of repeatedly freeze–thawing surimi. Significant differences among different concentrations of WPH samples under the same number of FT cycles are indicated by different lowercase letters (a–d). Significant differences among the same samples with different numbers of FT cycles are denoted by uppercase letters (A–C). Control: no additives in the sample; NWP: native whey protein isolate; WPH: whey protein hydrolysates; BHA: butylated hydroxyanisole.

**Figure 8 foods-12-04464-f008:**
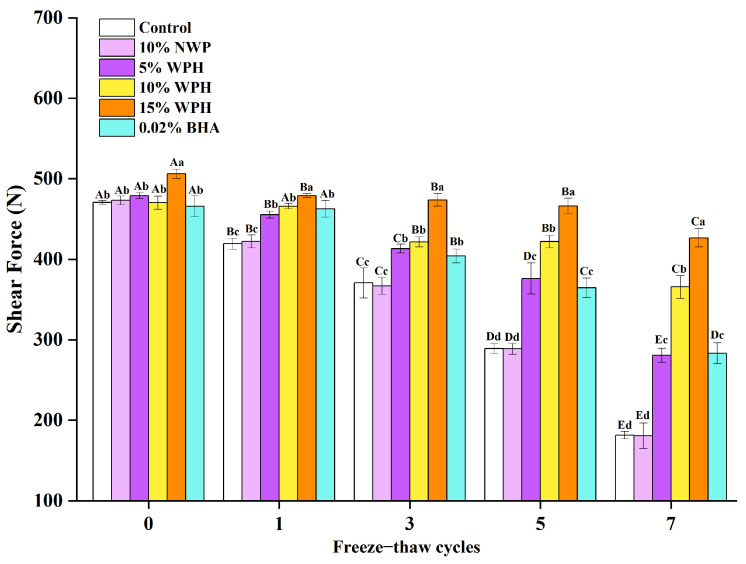
Effects of WPH on shear force stability of repeatedly freeze–thawing surimi. Significant differences among different concentrations of WPH samples under the same number of FT cycles are indicated by different lowercase letters (a–d). Significant differences among the same samples with different numbers of FT cycles are denoted by uppercase letters (A–E). Control: no additives in the sample; NWP: native whey protein isolate; WPH: whey protein hydrolysates; BHA: butylated hydroxyanisole.

## Data Availability

Data is contained within the article.

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
