# Peer review of "Protection of Whey Polypeptide on the Lipid Oxidation, Color, and Textural Stability of Frozen–Thawed Spanish Mackerel Surimi"

_foods, 2023, doi:10.3390/foods12244464_

Round 1
Reviewer 1 Report
Comments and Suggestions for Authors
L9-12: I wonder whey polypeptide was used as antioxidant or cryoprotectant? The high concentration up to 15% suggest its cryoprotecting effects in surimi and in this case the use of BHA as a synthetic antioxidant is ambiguous as it is supposed to use a cryoprotective substance such as sucrose, sorbitol, trehalose, as in L34 “Cryoprotectants must be added to stabilize the surimi's quality and texture” why not used a commercial cryoprotectant substance instead of BHA? Please explain.
When talk about antioxidant capacity in title, it means measuring endogenous muscle antioxidative protease such as SOD or catalase, as there is no evidence of such analysis, the title should be changed appropriately.
L14-15: Why higher antioxidant capacity of surimi is along with higher TVN? It was supposed to decrease in stabilized surimi. However, this is in contrast to Fig. 2 in which TVN is lower in treated surimi compared with the control during all FT cycles. Please explain.
L19: raw of cooked surimi?
L465: remove “A blank control group was also established.”
Reviewer 2 Report
Comments and Suggestions for Authors
The objective of the present study was to investigate the protective role of whey protein on surimi made of Spanish mackerel quality subjected to several freeze-thaw cycles.
The research is interesting and the authors made a number of analytical determinations to better highlight the effect of the additive.
However, I therefore have to point out some comments:
I suggest adding something about the composition or differences of Spanish mackerel when compared with other raw material used to elaborate surimi.
Keywords:
Please avoid to use those words already present in the title.
Material al methods:
I wonder what is the purpose of including the NWP group. I understand that the authors have included a negative control with no WPH added, a positive control with an antioxidant (BHA) and 3 WPH at three different concentrations. But why NWP? and why at 10%? Please, explain it.
Indeed, authors did not mention this group in the objectives of the investigation.
Remember that binomial nomenclature (like Scomberomorus niphonius or Spondias mombin L.) needs to be written in Italic and the first letter of genus must be written in upper case.
References:
I suggest avoiding lumping the references. Each reference should be discussed separately or deleted older. (See lines 28, 32, 37, 47, 55, 196, 219, 261, 335 and 439.
Some detailed comments are included directly in the manuscript attached.
